# Towards the Hydrophobization of Thermoplastic Starch Using Fatty Acid Starch Ester as Additive

**DOI:** 10.3390/molecules27196739

**Published:** 2022-10-10

**Authors:** Caroline Terrié, Angélique Mahieu, Vincent Lequart, Patrick Martin, Nathalie Leblanc, Nicolas Joly

**Affiliations:** 1Univ. Artois, UniLaSalle, ULR7519—Transformations & Agro-Ressources, Normandie Université, F-76130 Mont-Saint-Aignan, France; 2UniLaSalle—Ecole des Métiers de l’Environnement, Campus de Ker Lann, F-35170 Bruz, France; 3Univ. Artois, UniLaSalle, ULR7519—Unité Transformations & Agro-Ressources, F-62408 Béthune, France

**Keywords:** thermoplastic starch, fatty acid starch esters, hydrophobization, composition-properties relationship, dynamic vapor sorption (DVS), contact angle

## Abstract

To bring surface hydrophobicity to thermoplastic starch (TPS) materials for food packaging, fatty acid starch esters (FASE), specifically starch tri-laurate, were incorporated into TPS formulations. A total of three different ratios of FASE (2%, 5% and 10%) were added to the TPS formulation to evaluate the influence of FASE onto physico-chemical properties of TPS/FASE blends, i.e., surface hydrophobicity, dynamic vapor sorption (DVS), and tensile behaviors. Blending TPS with FASE leads to more hydrophobic materials, whatever the FASE ratio, with initially measured contact angles ranging from 90° for the 2%-FASE blend to 99° for the 10%-blend. FT-IR study of the material surface and inner core shows that FASE is mainly located at the material surface, justifying the increase of material surface hydrophobicity. Despite this surface hydrophobicity, blending TPS with FASE seems not to affect blend vapor sorption behavior. From a mechanical behavior perspective, the variability of tensile properties of starch-based materials with humidity rate is slightly reduced with increasing FASE ratio (a decrease of maximal stress of 10–30% was observed for FASE ratio 2% and 10%), leading to more ductile materials.

## 1. Introduction

Concern for the environment over several decades has led to extensive work aimed at replacing petrochemical-based plastic materials with renewable polymers, which, in general, provide environmental sustainability, unique performance, and economic benefits [1]. Starch, a polysaccharide extracted from various plants (e.g., corn, wheat, rice, potatoes), is one of the most widely studied biopolymers in this field due to its low cost, abundance, wide geographical distribution, and ease of growth. From a structural perspective, the two major macromolecules composing starch are amylose, a linear polymer consisting of α-1,4 linked D-glucose units, and amylopectin, a highly branched structure of short α-1,4 chains linked by α-1,6 bonds. Furthermore, starch can be processed via traditional polymer processing techniques, such as extrusion and injection molding, employing plasticizers [2,3,4,5] to obtain ThermoPlastic Starch (TPS). TPS has several advantages, such as total compostability and renewability of the resource [2], and shows a wide range of properties according to the starch botanical source, the plasticizer level, and the forming process used [6,7]. TPS-based materials could be particularly interesting for short-life-time food packaging [8] since their oxygen permeability is low when they contain a low amount of plasticizers [9] compared to most polyesters [10].

However, the hydrophilic nature of TPS has limited widespread use for industrial polymer applications since they require tailored properties such as mechanical integrity. Moreover, its water sensibility strongly influences both gas permeability and mechanical properties, which respectively increase and decrease in high-humidity environments [11].

To overcome these issues while maintaining biodegradability, one strategy consists of associating TPS with another biodegradable polymer [12]. Blends of TPS with several synthetic biodegradable polymers have been investigated in the literature, either with bio-based polymers such as poly(lactic acid) [13,14], polybutylene succinate adipate [15] or petroleum-based ones, e.g., polycaprolactone [9,16,17], poly(vinyl alcohol) [18,19], or poly(ethylene oxide) [20].

Chemical modification is another method to improve TPS properties [21,22,23]. Grafting of long hydrophobic chains onto starch can improve its water resistance, barrier properties and even its multilayer compatibility. It can also be an alternative to the use of plasticizers [1]. Several starch modifications have been investigated: cross-linking, esterification, etherification, oxidization, etc. [24]. 

Esterification of starch is among the several chemical modification efforts extensively reported in the literature. For instance, esterification with fatty acid derivatives, such as acid anhydrides or chlorides, is an interesting way to introduce both hydrophobicity and thermoplasticity in starch-based materials [25]. 

Starch modification with fatty acid derivatives has received substantial attention in recent times because of the fatty acid hydrophobicity and the ability to incorporate desired properties when grafted onto starch. Fatty acid starch esters (FASEs) properties may vary depending on the DS (degree of substitution: number of grafted fatty chains onto starch anhydroglucose units; maximum DS = 3) and fatty chain length. Studies have shown that fatty ester groups act as an internal plasticizer, leading to a decrease in the glass transition as the number and the length of fatty chains grafted onto starch are increased [21]. Moreover, a high DS may result in a thermoplastic and hydrophobic material [1].

Grafting of short-chain fatty acids (C2 to C5) onto starch has been studied by some groups [26,27]. Water vapor permeability was improved, but the resulting materials were brittle and required external plasticizers. Conversely, using medium (C6–C12) and long chain (C13–C21) fatty acids to modify starch could offer better processability and mechanical properties. However, the increase in chain length could result in limited substitution efficiency (low DS) due to steric hindrance and unfavorable side reactions [28].

Considering all these arguments, the present study aimed to overcome the moisture sensitivity of TPS by associating the thermoplastic extruded starch with a starch-based derivative as a hydrophobic additive. For this purpose, a fully substituted fatty acid starch ester (FASE) was used, namely starch tri-laurate. Starch tri-laurate was introduced at 2%, 5% and 10% in the formulation of the thermoplastic starch (wheat starch with 20% glycerol), and films were carried out by double extrusion (first extrusion for compounds, second extrusion for films). Films were subjected to several physico-chemical analyses, including thermogravimetric analyses, contact angle measurements, dynamic vapor sorption measurements and tensile tests, to evaluate their properties and the influence of FASE content on these characteristics.

## 2. Results

### 2.1. FASE Characterizations

FASE was first characterized by FT-IR spectroscopy, using the ATR method, to attest to the efficiency of reactions and confirm the grafting of the lauroyl fatty chain onto the starch backbone by an ester junction (Figure 1). 

To complete the chemical characterization of FASE, especially the determination of the degree of substitution (DS), FASE was analyzed by ^1^H NMR spectroscopy. The spectrum of FASE is depicted in Figure 2. 

### 2.2. Influence of FASEs Ratio on Properties of TPS/FASE Materials

#### 2.2.1. FT-IR Spectroscopy and Surface Analyses

Homogeneous TPS/FASE extruded films were successfully obtained by introducing FASE powder into a TPS formulation. Figure 3 shows the FTIR spectra of TPS-0, TPS-2, TPS-5, and TPS-10 at the surface using ATR equipment to confirm the incorporation of FASE in TPS materials.

FTIR analyses were performed on the surface and at 1/3rd of the thickness below the surface after cutting out the surface. Figure 4 and Figure 5 compare the FT-IR spectra at the core and the surfaces (upper and lower), respectively, for a TPS-5 film and a TPS-10 film.

#### 2.2.2. Hydrophobic Behavior and Kinetics of Water Sorption at the Surface of TPS/FASE Blends 

The hydrophobic behavior of TPS/FASE material surfaces was evaluated according to the FASE ratio using an Apollo Instruments OCA 20 contact angle apparatus at room temperature and ambient humidity by measuring the angle between a deionized water drop of 3 μL and the surface of tested material (Figure 6a). Moreover, a kinetic study of water sorption at the surface of TPS/FASE blends was also conducted by monitoring water contact angle evolution versus time (Figure 6b).

#### 2.2.3. Dynamic Vapor Sorption Measurements for TPS/FASE Blends

Dynamic vapor sorption measurements for TPS/FASE films are shown in Figure 7. All curves display similar shapes typical of starch-based materials [29]. 

#### 2.2.4. Mechanical Behavior of TPS/FASE Blends

The mechanical behaviors of the various TPS/FASE blends are presented in Table 1 as a function of the FASE ratio at different relative humidities. 

## 3. Discussion

### 3.1. FASE Synthesis and Characterization

FT-IR spectrum, depicted in Figure 1, is characteristic of a saturated fatty acid polysaccharide ester, with specific signals corresponding to ester carboxylic function (1740 cm^−1^), CH_2_ and CH_3_ symmetric and antisymmetric in-plan elongation (between 2800 and 2980 cm^−1^), and rocking vibration of linear CH_2_ sequence (720 cm^−1^). The other signals, mainly between 900 and 1300 cm^−1^, correspond to the starch backbone. The absence of the hydroxyl group-specific band at 3300–3400 cm^−1^ indicates that the grafting of fatty chains onto the starch skeleton was almost completely achieved. 

Starch acylation was also confirmed by ^1^H NMR spectroscopy. ^1^H NMR spectrum (Figure 2) attested to the presence of characteristic signals of fatty chain protons from 0.89 to 2.34 ppm and of the starch backbone from 3.00 to 5.50 ppm (H_starch_, carbohydrate protons). The degree of substitution (DS) of FASE was determined using the ^1^H NMR integrations method described in the literature for fatty acid cellulose esters [30]. DS was estimated at 2.95 ± 0.1, confirming the supposition emitted about FT-IR analysis that starch hydroxyl groups were totally grafted by fatty acid chains. Furthermore, ^1^H NMR was also used to confirm the lack of residual synthetic products, such as pyridine used during synthesis and fatty acid methyl ester-induced during reaction purification. No signal was observed either at 3.68 ppm for the methyl ester group or higher than 6.5 ppm for pyridine protons. To conclude, the FASE that will be used further as a hydrophobic additive in TPS formulation is pure and can be considered a fully substituted starch (DS = 3).

### 3.2. Properties of TPS/FASE Materials according to FASE Ratios

To evaluate the influence of FASE and FASE ratio on the surface properties of formulated TPS materials, we have performed surface analyses, either for chemical and physical properties, on samples stored at 23°C and 50% RH. The first analyses concerned were FT-IR spectroscopy (Figure 3). Increasing the FASE rate led to an increase in the intensity of the C-H alkyl bonds characteristic signals at 2800–2900 cm^−1^ (-C-H antisymmetric and symmetric stretching of -CH_2_- and -CH_3_), corresponding to the presence of fatty long chains. The appearance of new bands around 1740 cm^−1^ attributed to the stretching of carbonyl ester groups was observed. This result confirmed the efficiency of introducing FASE into TPS by extrusion blending since FASE can be found at least at the surface of TPS/FASE-formulated materials. Consequently, contact angle measurement has shown that every TPS/FASE surface is hydrophobic, compared to hydrophilic TPS-0 one, with contact angles higher than 90°, whatever the FASE ratio. 

The study of the contact angle evolution as a function of time allowed us to check the effective hydrophobicity of the surface of TPS/FASE blends (Figure 6b). First, for TPS-0 results, a notable decrease in contact angles is observed, meaning that water drop was adsorbed in the support. Conversely, for each ratio of FASE, we can observe a slight decrease in the contact angle as a function of time. This slight decrease can be attributed to the evaporation of the water and not to the adsorption of the water in the support. After 2 min, despite the slight decrease in contact angles of TPS/FASE surfaces according to time, the TPS-2 surface is still more hydrophobic than the TPS-0 one (84° compared to 40° for TPS-0). Moreover, TPS-5 and TPS-10 surfaces remained hydrophobic with contact angle values of 90° and 96°, respectively.

To evaluate if FASE is homogeneously distributed into TPS, FT-IR analyses (Figure 4 et Figure 5) were performed at the core and the surfaces (upper and lower), respectively, for a TPS-5 film and a TPS-10 film. In both cases, the intensity of the bands at 2800–2900 cm^−1^ and 1740 cm^−1^, characteristics for FASE, is stronger at the surface than at the core. This suggests that FASE was located preferentially towards the film surface, whatever the FASE ratio was. This could also explain the surface hydrophobicity of TPS/FASE blends *versus* TPS-0.

Because of the heterogeneity of FASE distribution among TPS/FASE blend and their surface hydrophobicity, dynamic vapor sorption measurements for TPS/FASE films were performed to evaluate the cohesion of TPS/FASE blends and their behavior against humidity. According to the results depicted in Figure 7, no significant difference is observed between the samples, despite their differences in surface hydrophobicity. However, as the relative humidity increased, a swelling of the samples occurred, inducing a loss of the protective layer on the edges. Perhaps the water sorption by the edges of the films prevented the differentiation of TPS/FASEs sorption behavior.

Finally, TPS/FASE films were submitted to tensile tests to evaluate the influence of FASE amount on mechanical behavior at different relative humidities (Table 1). The general trend is a decrease in maximum stress and modulus as the FASE ratio increases. The values at 35% RH are much higher than the others. At this relative humidity, the samples are dry and very brittle, even if they become more “usable” when the FASE ratio increases. The results obtained at 50%, 59% and 66% RH are more representative of standard conditions of use of the materials. We can see that the maximal stress and elasticity modulus decrease with the increase of the FASE ratio for the materials conditioned at 50% RH, whereas the evolution of both these properties is less significant for materials conditioned at 59% and 66% RH. Finally, the material with a 10% FASE ratio presents much less variability in its mechanical properties as a function of humidity rate compared to TPS-0. 

The mechanical properties of starch-based materials depend strongly on moisture content, which is one of their major drawbacks. The variation of maximum stress and elastic modulus of TPS/FASEs with relative humidity is reduced when the FASE ratio increases but remains significant if we consider materials stored at 35% RH. The samples used for the tensile tests were cut in the center of the extruded films and then stored in humidity-controlled chambers. Since FASEs are located preferentially towards the film surface, the sides of the samples that have been cut out are expected to be more hydrophilic than the upper and lower surfaces of the sample. Even if the surface area of the cut sides is small compared to the total surface area of the sample, this could contribute to the variation of the tensile properties with humidity rate. Another forming technique for the tensile samples could be considered, such as injection, to have an external layer of FASE on the entire material surface.

## 4. Materials and Methods

### 4.1. Materials 

Unmodified wheat starch and glycerol were obtained from Sigma Aldrich (France). Wheat starch contains 26% amylose (Mw ≈ 1,5.10^6^ g.mol^−1^) [31] and 74% amylopectin. Mw of amylopectin was estimated to be around 1.10^9^ g.mol^−1^ [32]. 

For the synthesis of fatty acid starch ester (FASE), all reagents were stored at room temperature and used without further purification: pyridine (99%, Acros, Paris, France); lauroyl chloride (LCl, 98%, Aldrich, Paris, France) chloroform (≥99%, Carlo Erba, Paris, France); methanol (≥99%, Carlo Erba, Paris, France). Deuterated chloroform used for NMR analyses was purchased from Sigma Aldrich and stored at 4 °C.

### 4.2. Synthesis of FASE

FASE was synthesized using a procedure adapted from literature [33] to obtain fully substituted starch laurate with a good conversion. In a typical experiment, 20 g of starch (123 mmol of anhydroglucose unit) was suspended in 200 mL of pyridine at room temperature. Next, lauroyl chloride (738 mmol, 180 mL, 6 equiv./anhydroglucose unit) was added, and the reaction mixture was subsequently heated to a constant 110 °C and stirred for 1 h. After cooling to room temperature, the mixture was added to 500 mL methanol. The brownish raw product was recovered by filtration and dissolved in chloroform. Starch laurate was precipitated by being added dropwise to methanol. Following subsequent washing steps with methanol, the product was dried in a vacuum (85 °C, 16 h).

### 4.3. FASE-TPS Blending Process

In the first step, starch was blended with glycerol in a thermo-regulated turbo-mixer and heated in an oven at a constant 170 °C for 45 min, allowing glycerol diffusion into the starch granules. Next, FASE powder was added, and the humidity rate was adjusted to 10% *w*/*w*. The resulting mixture was extruded and granulated in a single screw extruder (SCAMEX, Paris, France) equipped with two heating zones regulated at 100 °C, then 120 °C around the screw and 110 °C for the die, at a screw speed of 50 rpm.

The initial compositions of the samples prepared are listed in Table 2. 20% glycerol is the maximum rate leading to starch plasticization without phase separation between starch and glycerol [34,35].

In a second step, these granules were extruded through a slit die and calendered to obtain films (average thickness: 1.5 ± 0.3 mm). Temperatures on the extruder were 110 °C for both heating zones and 90 °C for the die, at a screw speed of 50 rpm. Films were stored in controlled chambers at 55% relative humidity (RH).

### 4.4. Characterization of FASE and Blends

#### 4.4.1. FT-IR Analyses of FASE and TPS/FASE Blends

FASE synthesized during this study was characterized after casting by Fourier Transformed InfraRed (FT-IR) spectroscopy using an Agilent Cary 630 apparatus equipped with an ATR accessory to probe the efficiency of reactions. A series of 64 scans were collected for each measurement over the 4000 to 650 cm^−1^ spectral range with a 4 cm^−1^ resolution.

ATR–FT-IR spectroscopy was also used to characterize TPS/FASE blends. Measurements were performed separately on the film surface or the film core. The “core” measurement was executed at 1/3rd of the thickness below the surface. As for FASE analysis, a series of 64 scans were collected for each measurement over the 4000 to 650 cm^−1^ spectral range with a 4 cm^−1^ resolution. 

#### 4.4.2. ^1^H NMR Analyses of FASE

^1^H NMR spectra were performed in CDCl_3_ using a Bruker DRX-300 Spectrometer (operating at 300 MHz) to assess purity (lack of methyl laurate and pyridine-specific signals) and confirm the degree of substitution (DS) of FASE by an integration method described and proven in previous works for fatty acid cellulose esters [21,36,37]. In a typical experiment, 15 mg of FASE was dissolvent in 0.5 mL of CDCl_3_ prior to analysis.

#### 4.4.3. Surface Behaviors Versus Water of TPS/FASE Blends

The influence of FASE ratios on TPS/FASE blends surface behaviors versus water was evaluated by contact angle measurements using an Apollo Instruments OCA 20 contact angle apparatus. The sessile drop method was applied, with a deionized water drop of 3 μL, at room temperature and ambient humidity. All contact angles were measured on both sides of the drop by the ellipse-fitting calculation method. Contact angles, i.e., right angle θ_R_, left angle θ_L_ and average angle θ_moy_, reported in this work are an average of the values obtained for a minimum of four points on the sample surface. The evolution of contact angles versus time was also determined by measuring both right and left contact angles for 2 min.

#### 4.4.4. Dynamic Vapor Sorption (DVS) of TPS/FASE Blends

Sorption isotherms were measured with the SPSx Vapor Sorption Analyser (ProUmid, Germany). This device allows monitoring of the mass uptake and the sorption kinetics of one sample accurately using a microbalance and precise control of both temperature and humidity. The sample (~1g) is exposed to increasing humidity from 35% to 80%, in 15% humidity steps, then to 90% and then back to 35% following the same steps. The testing temperature is 25 °C. The equilibrium mass at each step is determined when mass variation is less than 0.001% or after 15 min. The edges of the samples were protected with a hot melt adhesive.

#### 4.4.5. Tensile Behavior of TPS/FASE Blends

Tensile tests were carried out using a universal testing machine (TA.XTplus Texture Analyzer, France) equipped with a load cell of 500 N and following the international standard ASTMD882. Average values and standard deviations of five different standard dumbbell specimens were reported. The samples were taken in the center of the extruded films and the extrusion drawn direction. The sample geometry is a standard traction specimen [38]. Prior to analysis, the samples were conditioned in controlled chambers at different relative humidities using saturated salt solutions, and the tests were performed at room temperature and humidity under atmospheric pressure. The thickness of each sample was determined at four different positions. The films were placed in the grips and stretched at 6 mm.min^−1^ until breakage. The average parameters, E (elastic modulus, MPa), σ_m_ (tensile strength at break, MPa) and ε (deformation at the fracture point, expressed in percentage), were calculated.

## 5. Conclusions

The starch modification process developed allows for the successful synthesis of hydrophobic starch tri-laurate, and this product can be blended with thermoplastic starch to obtain homogeneous films. This association leads to more hydrophobic films that could be used as food packaging materials.

FASEs are located at the surface of the final material, and surface hydrophobicity increases with the FASE ratio. However, despite this surface hydrophobicity, no effect on the vapor sorption behavior was evidenced. The variation of mechanical properties with humidity rate, typical of starch-based materials, is slightly reduced with increasing starch tri-laurate ratio.

A different forming technique for the characterization samples or the whole material should be tested in further studies to obtain a material with FASE distributed on the whole surface. This seems to be the condition to observe a significant improvement in the water sensitivity of the starch-based materials thanks to FASE.

It is planned to take advantage of the compatibility of starch tri-laurate with plasticized starch, associated with the migration of FASE towards the surface, in a multilayer material made of a starch-based layer with a hydrophobic polymer. It can also be envisaged in further studies to synthesize the starch tri-laurate by reactive extrusion as a continuous production method.

## Figures and Tables

**Figure 1 molecules-27-06739-f001:**
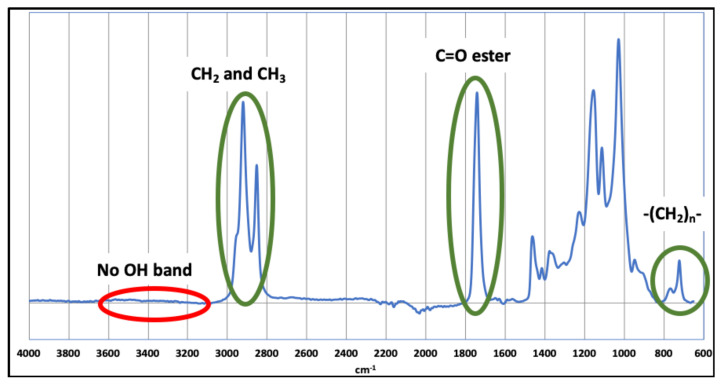
FT-IR spectrum of FASE.

**Figure 2 molecules-27-06739-f002:**
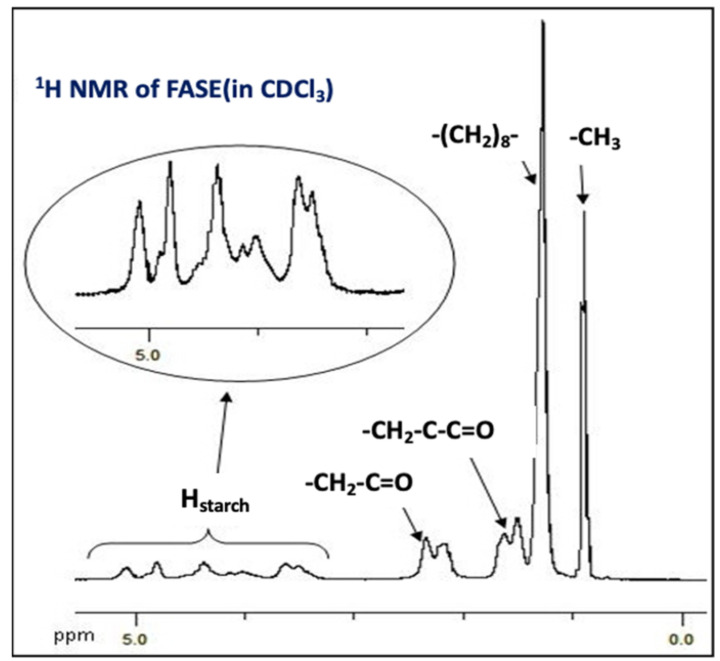
^1^H NMR spectrum of FASE (in CDCl_3_).

**Figure 3 molecules-27-06739-f003:**
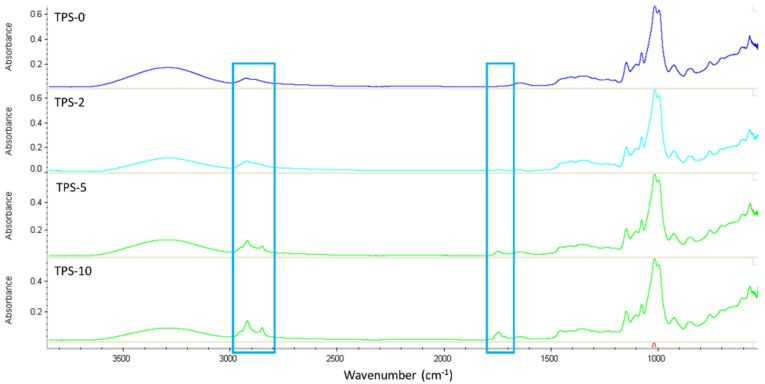
FTIR spectra of TPS/FASE blends (from top to bottom: TPS-0, TPS-2, TPS-5, and TPS-10.

**Figure 4 molecules-27-06739-f004:**
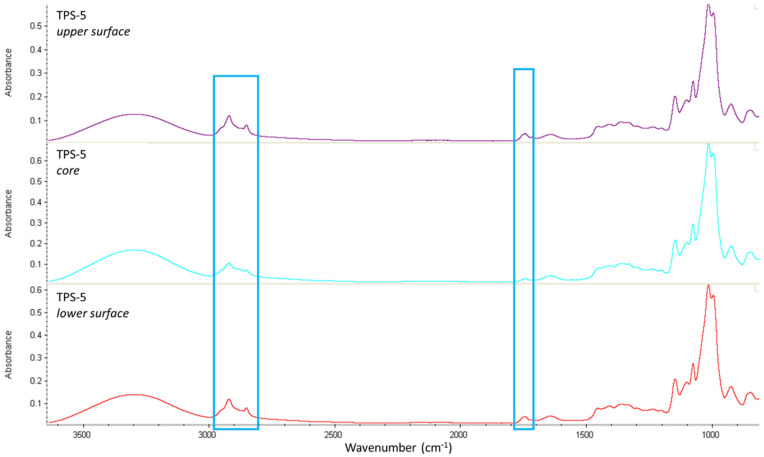
FTIR spectra for TPS-5 (upper surface, core and lower surface).

**Figure 5 molecules-27-06739-f005:**
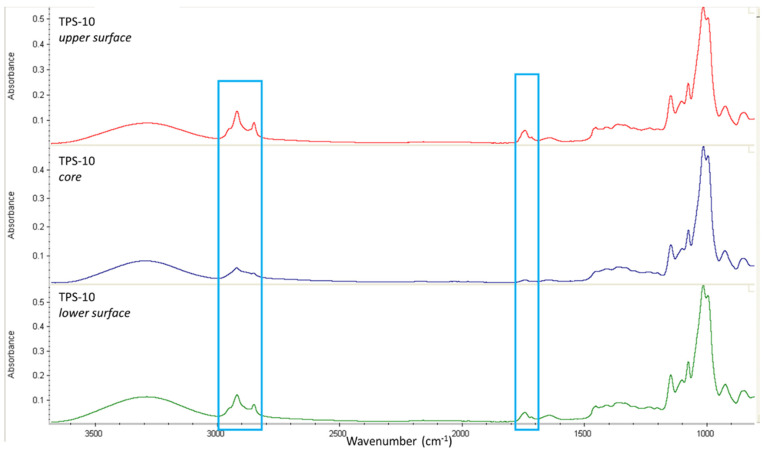
FTIR spectra for TPS-10 (upper surface, core and lower surface).

**Figure 6 molecules-27-06739-f006:**
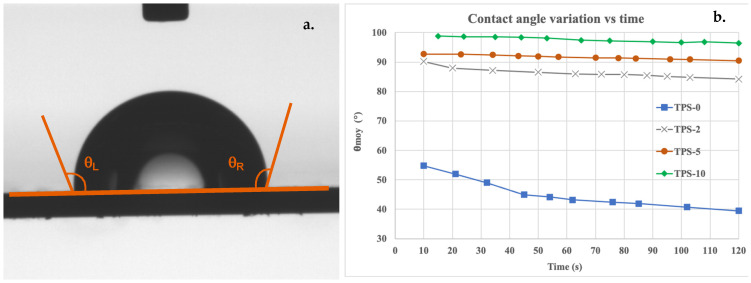
Surface contact angle picture for TPS-10 (**a**) and kinetic of water sorption according to FASE ratio in TPS/FASE-ratio (**b**).

**Figure 7 molecules-27-06739-f007:**
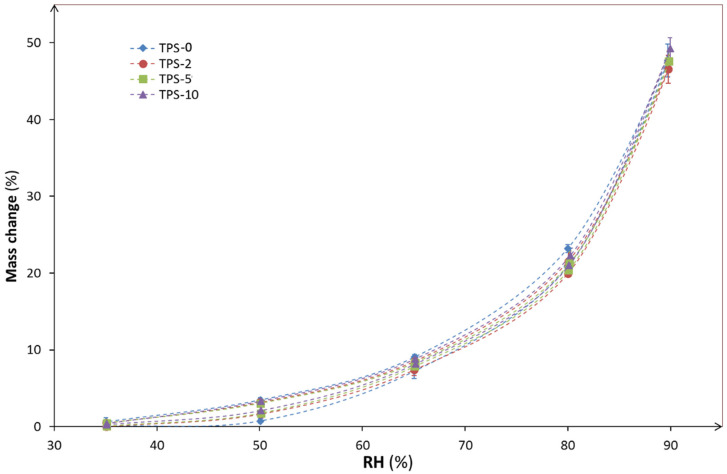
Dynamic vapor sorption curves for TPS/FASEs blends.

**Table 1 molecules-27-06739-t001:** Mechanical properties of TPS/FASE blends (Elastic Modulus; Maximum stress; Maximum strain) as a function of the FASE ratio at different relative humidities.

Name	RelativeHumidity	Elastic Modulus (MPa)		Maximal Stress (MPa)		Maximal Strain (%)	
		E	+/−	σ_R_	+/−	ε	+/−
TPS-0	35%	4065	247	21.3	0.4	9	1
	50%	368	89	5.4	0.1	154	11
	59%	69	11	2.4	0.2	191	8
	66%	141	12	3.3	0.1	84	14
TPS-2	35%	4558	229	21.4	2	6	1
	50%	348	109	5.2	0.3	142	5
	59%	69	6	2.8	0.1	179	10
	66%	152	8	3.4	0.1	81	2
TPS-5	35%	3512	215	15.1	1.1	5	1
	50%	301	83	4.4	0.5	140	12
	59%	75	7	2.6	0.2	158	11
	66%	122	6	2.7	0.1	66	2
TPS-10	35%	2184	406	12.6	0.4	15	2
	50%	79	11	2.9	0.2	158	9
	59%	65	22	2.5	0.2	96	6
	66%	116	15	2.3	0.3	47	6

**Table 2 molecules-27-06739-t002:** Initial composition (% *w*/*w*) of the samples.

Name	Native Starch	Glycerol	FASE
TPS-0	80	20	0
TPS-2	78	20	2
TPS-5	75	20	5
TPS-10	70	20	10

## Data Availability

Not applicable.

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
