# Peer review of "Towards the Hydrophobization of Thermoplastic Starch Using Fatty Acid Starch Ester as Additive"

_molecules, 2022, doi:10.3390/molecules27196739_

Round 1

Reviewer 1 Report

The article examine the hydrophobization of thermoplastic starch by blending TPS with starch laurate. Authors confirmed substitution in starch laurate with FT-IR and NMR.

The methodology is described correctly. The results are clear and well explained. However there are some inconsistencies in the text

The numbering of the Figures, starting with Figure 4 on page 6, is inconsistent.

Line 127 – reference to dynamic vapor sorption measurements figure should be updated

Line 130 - Number of Figure 4 should be figure 7

Line 132 - reference to mechanical measurements figs.  should be updated

Line 134 - Number of Figure 5 should be figure 8

Line 182 – references to figs. should be updated

Based on the high quality of the manuscript I suggest it to be accepted after minor revision.

Author Response

Dear reviewer

Thank you for your nice comments. To answer your remarks :

"The numbering of the Figures, starting with Figure 4 on page 6, is inconsistent."

This was modified in the manuscript

"Line 127 – reference to dynamic vapor sorption measurements figure should be updated"

This was modified in the manuscript

Line 130 - Number of Figure 4 should be figure 7

This was modified in the manuscript

Line 132 - reference to mechanical measurements figs.  should be updated

This was done in the manuscript

Line 134 - Number of Figure 5 should be figure 8

This was modified in the manuscript

Line 182 – references to figs. should be updated

This was modified in the manuscript

Reviewer 2 Report

In general, the subject treated in the article "Towards the hydrophobization of thermoplastic starch using fatty acid starch ester as additive" by C. Terrie, et al. is very interesting. The article shows novel results in the field of thermoplastic starch. However, the article needs minor changes to be published in Molecules.

The following is a list of observations:

1. Abstract, line 20: how much is reduced the tensile properties?

2. Keywords: add the terms “dynamic vapour sorption (DVS)”, “contact angle”

3. In Figure 6 identify the components involved, i.e. the water droplet and the test material.

4. In the DVS graphs (Figure 4), the 59% RH values were not recorded; any special reason for not showing them?; since in the mechanical properties the values for 59% RH do appear.

5. Figures 4a and 4b are sufficient to show the results of mechanical properties; adding a "zoom" confuses the reader.

6. In line 188 the term "TPS/FASE blend" appears, different from the one used throughout the text, as "TPS-FASE blends", it is more convenient to use the notation xx/xx instead of xx-xx".

7.  In line 197 correct “(Figure 8)” by “(Figure 5)”.

8. In Table 1, it is convenient to write TPS-0, TPS-2, TPS-5 and TPS-10, without the % symbol.

Author Response

Dear reviewer

Thank you for your help in the improvement of this manuscript. Here are the answers to your observations

  1. Abstract, line 20: how much is reduced the tensile properties?

The answer was added in the abstract : “(a decrease of maximal stress of 10-30% was observed for FASE ratio 2% and 10%),”

  1. Keywords: add the terms “dynamic vapour sorption (DVS)”, “contact angle”

terms were added in keyword section

  1. In Figure 6 identify the components involved, i.e. the water droplet and the test material.

Photo was identified in the figure legend : “Surface contact angle picture for TPS-10”

  1. In the DVS graphs (Figure 4), the 59% RH values were not recorded; any special reason for not showing them?; since in the mechanical properties the values for 59% RH do appear.

The values of relative humidity for the DVS measurements and for tensile experiments are different only for practical reasons. For DVS measurements the RH values are set by the equipment (at 35, 50, 65, 80 and 90%); for tensile experiments the samples were conditioned in controlled chambers at different relative humidities using saturated salt solutions, which allowed us to obtain 35, 50, 59 and 66% RH. However, the sorption curves can be used to assess the water absorption at 59% RH.

  1. Figures 4a and 4b are sufficient to show the results of mechanical properties; adding a "zoom" confuses the reader.

Figure 4 was converted into a table, as recommanded by reviewer3

  1. In line 188 the term "TPS/FASE blend" appears, different from the one used throughout the text, as "TPS-FASE blends", it is more convenient to use the notation xx/xx instead of xx-xx".

Modification was done all along the manuscript

  1. In line 197 correct “(Figure 8)” by “(Figure 5)”.

corrected

  1. In Table 1, it is convenient to write TPS-0, TPS-2, TPS-5 and TPS-10, without the % symbol.

Modification was done in table 1 and all along the manuscript

Reviewer 3 Report

Minor spell check is required for the contribution.

Reconsider to modify sub-section 2.2.4 title to adapt to its content. Figure 5 should be reconsidered. Please consider including plots of stress-strain dependencies, and/or table on mechanical properties (e.g., elastic moduli, tensile strength, elongation at break, etc.). 

Supplementary information on characterization methods (see section 4), conditions and data processing should be inserted to ensure reproducibility.

Please reconsider statement with page 8, rows 216-218 to be included in conclusions or omitted proven no proof was brought forth.    

Author Response

Dear reviewer

Many thanks for your review and your remarks. Here are the answers to your recommendations

"Minor spell check is required for the contribution."

the whole Manuscript was read twice and spell mistakes were corrected. thank you

"Reconsider to modify sub-section 2.2.4 title to adapt to its content. Figure 5 should be reconsidered. Please consider including plots of stress-strain dependencies, and/or table on mechanical properties (e.g., elastic moduli, tensile strength, elongation at break, etc.). "

Title of subsection  2.2.4 was changed. Figure 8 was converted into a table reporting corresponding data (elastic moduli, max stress, max strain)

"Supplementary information on characterization methods (see section 4), conditions and data processing should be inserted to ensure reproducibility."

Supplementary information were added, especially for contact angle measurement and NMR spectroscopy

"Please reconsider statement with page 8, rows 216-218 to be included in conclusions or omitted proven no proof was brought forth.  " 

The concerned sentence was reconsidered : “Since FASE are located preferentially towards the film surface, the sides of the samples that have been cut out are expected to be more hydrophilic than the upper and lower surfaces of the sample. Even if the surface area of the cut sides is small compared to the total surface area of the sample, this could contribute to the variation of the tensile properties with humidity rate. Another forming technique for the tensile samples could be considered, such as injection, in order to have an external layer of FASE on the whole surface of the material.”

Round 2

Reviewer 3 Report

Congratulation on your contribution!